# EMOVA : Empowering Language Models to See, Hear and Speak with Vivid Emotions

## Abstract

GPT-4o, an omni-modal model that enables vocal conversations with diverse emotions and tones, marks a milestone for omni-modal foundation models. However, empowering Large Language Models to perceive and generate images, texts, and speeches end-to-end with publicly available data remains challenging in the open-source community. Existing vision-language models rely on external tools for the speech processing, while speech-language models still suffer from limited or even without vision-understanding abilities. To address this gap, we propose **EMOVA** (**EM**-otionally **O**mni-present **V**oice **A**ssistant), to enable Large Language Models with end-to-end speech capabilities while maintaining the leading vision-language performance. With a *semantic-acoustic disentangled* speech tokenizer, we notice surprisingly that omni-modal alignment can further enhance vision-language and speech abilities compared with the corresponding bi-modal aligned counterparts. Moreover, a lightweight style module is proposed for flexible speech style controls (*e.g.*, emotions and pitches). For the first time, **EMOVA** achieves state-of-the-art performance on both the vision-language and speech benchmarks, and meanwhile, supporting omni-modal emotional spoken dialogue. Demos are available in the project page: https://emova-anonymous.github.io/.

## 1 Introduction

OpenAI GPT-4o (OpenAI, 2024), a new milestone for omni-modal foundation models, has rekindled people's attentions on intelligent assistants that can *see* (*i.e.*, perceiving fine-grained visual inputs), *hear* (*i.e.*, understanding vocal instructions) and *speak* (*i.e.*, generating vocal responses) simultaneously. Most existing Multi-modal Large Language Models (MLLMs) focus on two modalities only, either vision-language (Bai et al., 2023; Li et al., 2024a) or speech-language (Chu et al., 2024; Xie & Wu, 2024), demonstrating severe demands for omni-modal models with visual, language and speech abilities. How to effectively empower Large Language Models (LLMs) to process omni-modal data in an end-to-end manner remains an open question.

Existing omni-modal LLMs (Chen et al., 2024b; Fu et al., 2024b) generally build upon Vision LLMs and integrate the speech modality by adopting a speech encoder like Whisper (Radford et al., 2023), which extracts **continuous** features from speech, similar to how images are processed, and enables speech understanding. However, these models still rely on external Text-to-Speech (TTS) tools for generating speech responses, limiting their ability to support real-time interactions. AnyGPT (Zhan et al., 2024), instead, opts for a fully **discretization** manner, which first discretizes all data modalities (*i.e.*, images, texts, and speeches), followed by omni-modal auto-regressive modeling. This enables AnyGPT to handle multiple modalities with a **unified end-to-end** framework, facilitating **real-time interactions** with the help of **streaming decoding**. However, the discrete vision tokenizer adopted by AnyGPT struggles to capture visual details, especially for high-resolution images, making it far behind its continuous counterparts on vision-language benchmarks. Moreover, none of the existing works explore speech style controls (*e.g.*, emotions and pitches) with LLMs. Therefore, our question arises: *How to build an end-to-end omni-modal LLM enabling spoken dialogue with vivid emotions while maintaining state-of-the-art vision-language performance?*

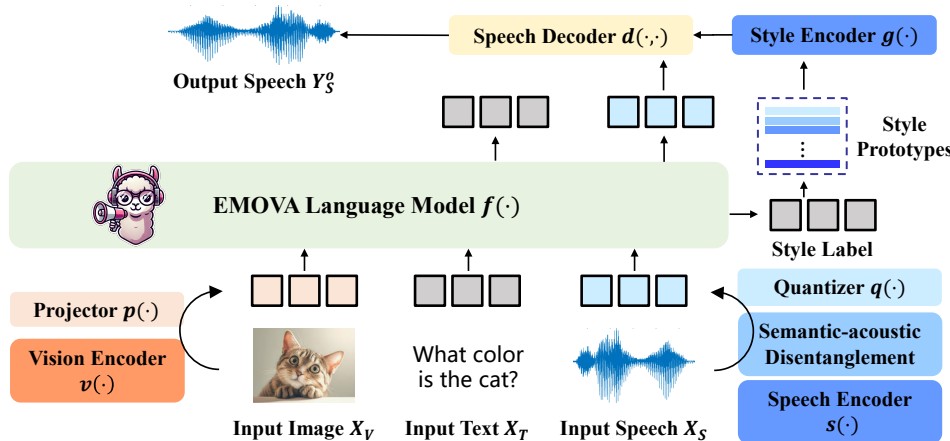

Figure 1: **Model architecture of EMOVA.** The vision encoder extracts continuous visual features, which are projected into the text embedding space as visual tokens, while the input speech is encoded and quantized into discrete units. Given the omni-modal inputs, **EMOVA** can generate both textual and speech responses with vivid emotional control. Check Sec. 3 for more details.

In this paper, we propose **EMOVA** (**EM**otionally **O**mni-present **V**oice **A**ssistant), a novel end-to-end omni-modal LLM with state-of-the-art vision-language and speech capabilities while supporting emotional spoken dialogue. Fig. 1 shows an overview of the model framework. A *continuous* vision encoder captures the fine-grained visual details, while the *discrete* speech tokenizer and detokenizer enable the end-to-end speech understanding and generation. Specifically, the speech-to-unit (S2U) tokenizer converts the input speech waveforms into discrete speech units as LLM inputs, while the unit-to-speech (U2S) detokenizer reconstructs the speech waveforms from the LLM's output speech units. To seamlessly integrate the speech modality with LLMs, we meticulously design a **semantic-acoustic disentangled** speech tokenizer to decouple the semantic contents and acoustic styles of the input speeches (Tao et al., 2024), where 1) *semantic content* (*i.e.*, what it says) captures the semantic meanings of input speeches, which is finally discretized and aligned with LLMs, while 2) *acoustic style* (*i.e.*, how it says) captures the diverse speech styles (*e.g.*, emotions and pitches). Utilizing the semantic-acoustic disentanglement of our speech tokenizer, we further introduce a lightweight style module to support spoken dialogue with vivid emotions and pitches. As in Sec. 4.1, this innovative disentanglement design better facilitates the modality alignment between texts and speeches while maintaining flexibility for diverse speech style controllability and personalization.

With the end-to-end omni-modal architecture of **EMOVA**, we empirically demonstrate that publicly available bi-modal image-text and speech-text data are sufficient for omni-modal alignment, utilizing the text modality as a bridge. This eliminates the need for omni-modal data (*i.e.*, image-text-speech), which is usually scarce. Surprisingly, we find that omni-modal alignment can further improve both vision-language and speech capabilities through joint optimization, even when compared with their bi-modal aligned counterparts. Finally, only a small amount of mixed-modality samples are required to teach the model to respond in the desired format. For the first time, **EMOVA** achieves state-of-the-art performance on both vision-language and speech benchmarks (see Table 1 for comparisons).

The main contributions of this work contain three parts:

1. We propose **EMOVA**, a novel end-to-end omni-modal LLM that can see, hear and speak. We use a continuous vision encoder and a semantic-acoustic disentangled speech tokenizer for seamless omni-modal alignment and diverse speech style controllability.

2. We introduce an efficient text-centric omni-modal alignment which can further improve the vision-language and speech capabilities, even compared with the corresponding bi-modal aligned counterparts (*i.e.*, image-text only and speech-text only alignment).

3. For the first time, our **EMOVA** achieve state-of-the-art comparable performance on both the vision-language and speech benchmarks simultaneously, while supporting flexible spoken dialogues with vivid emotions.

Table 1: **Comparison among Multi-modal Large Language Models.** Our **EMOVA** is the very first unified Omni-modal Large Language Model capable of emotional spoken dialogue with state-of-the-art vision-language and speech capabilities simultaneously.

| | Method | Visual | Text | Speech Understand | Speech Generation | Emotional |
|---|---|---|---|---|---|---|
| Vision | LLaVA | ✓ | ✓ | ✗ | ✗ | ✗ |
| | Intern-VL | ✓ | ✓ | ✗ | ✗ | ✗ |
| Speech | Qwen-Audio | ✗ | ✓ | ✓ | ✗ | ✗ |
| | Mini-Omni | ✗ | ✓ | ✓ | ✓ | ✗ |
| | LLaMA-Omni | ✗ | ✓ | ✓ | ✓ | ✗ |
| Omni | Intern-Omni | ✓ | ✓ | ✓ | ✗ | ✗ |
| | VITA | ✓ | ✓ | ✓ | ✗ | ✗ |
| | Any-GPT | ✓ | ✓ | ✓ | ✓ | ✗ |
| | **EMOVA (ours)** | ✓ | ✓ | ✓ | ✓ | ✓ |

## 2 RELATED WORK

**Vision Large Language Models** (VLLMs) integrate vision modality into Large Language Models (LLMs) (Touvron et al., 2023; Chen et al., 2023b), enabling advanced understanding and reasoning over visual instructions (Liu et al., 2024b; Bai et al., 2023; Gou et al., 2023; 2024). Recent efforts in VLLMs can be broadly categorized into three directions, including 1) *Vision encoders* (Oquab et al., 2023; Chen et al., 2021b; 2023a) are enhanced and aggregated for robust representations (Lin et al., 2023; Li et al., 2024b; Tong et al., 2024). 2) *High-resolution* methods are proposed to overcome the fixed resolution of pre-trained vision encoders (e.g., $336 \times 336$ for CLIP (Radford et al., 2021)), empowering LLMs to perceive fine-grained visual information (Liu et al., 2024a; Dong et al., 2024; Huang et al., 2024; Luo et al., 2024). 3) *High-quality instruction data* is essential for the VLLMs to generate accurate and well-formed responses following instructions (Laurençon et al., 2024; Li et al., 2024a; Chen et al., 2024b). In this paper, besides achieving state-of-the-art vision-language performance, we further introduce speech understanding and generating capabilities into our **EMOVA**.

**Speech Large Language Models** (SLLMs) empower speech interaction with LLMs. *Continuous SLLMs* (Wu et al., 2023; Chu et al., 2024) utilize the speech encoders (Radford et al., 2023) to extract continuous speech embeddings for LLM, which, however, only support speech understanding, relying on external TTS modules for speech generation, and therefore, hampering real-time interaction. *Discrete SLLMs* (Zhang et al., 2023a), instead, first discretize speech signals with speech tokenizers, followed by auto-regressive modeling. Recent works (Fang et al., 2024; Xie & Wu, 2024) further combine the continuous speech encoders with the discrete speech tokenizers for better performance. Although effective, none of the existing works explore speech style controllability in SLLMs (*e.g.*, genders, emotions, and pitches), which is essential for real-life spoken dialogue.

**Omni-modal Large Language Models** support visual, text, and speech capabilities with a unified architecture simultaneously. Similar to the continuous SLLMs, InternOmni (Chen et al., 2024b) and VITA (Fu et al., 2024b) connect a speech encoder with VLLMs, supporting speech understanding only. Instead, AnyGPT (Zhan et al., 2024) proposes a unified architecture to discretize and conduct auto-regressive modeling for image, text, and audio simultaneously, which, however, suffers from inevitable information loss brought by discretization, especially for the high-resolution visual inputs. In this work, we propose **EMOVA**, the very first unified Omni-modal Large Language Models with state-of-the-art vision-language and speech performance at the same time.

## 3 ARCHITECTURE

### 3.1 FORMULATION

Denote the Large Language Model (LLM) as $f(\cdot)$ and the text, visual and speech inputs as $\mathbf{X}_T$, $\mathbf{X}_V$ and $\mathbf{X}_S$, respectively. $\mathbf{X}_T$ is converted to discrete tokens $\mathbf{U}_T$ via a text tokenizer (Gage, 1994), while the visual input $\mathbf{X}_V$ is first encoded with a vision encoder $v(\cdot)$ as $\mathbf{E}_V = v(\mathbf{X}_V)$, and then projected into the text embedding space with a projector $p(\cdot)$ as $\mathbf{H}_V = p(\mathbf{E}_V)$. As for the speech input $\mathbf{X}_S$, a

*Speech-to-Unit* (S2U) procedure is required. Specifically, $\mathbf{X}_S$ first passes through a speech encoder $s(\cdot)$ as $\mathbf{E}_S = s(\mathbf{X}_S)$, which is then discretized by the quantizer $q(\cdot)$ as $\mathbf{U}_S = q(\mathbf{E}_S)$. The LLM $f(\cdot)$ is then trained to compute the joint probability of the output text and speech units $\mathbf{U}_T^o, \mathbf{U}_S^o$ as

$$\mathbb{P}(\mathbf{U}_T^o, \mathbf{U}_S^o | \mathbf{U}_T, \mathbf{U}_S, \mathbf{H}_V) = \prod_{i=1}^{L} \mathbb{P}(\boldsymbol{x}_i | \mathbf{U}_{T,<i}^o, \mathbf{U}_{S,<i}^o, \mathbf{U}_T, \mathbf{U}_S, \mathbf{H}_V), \tag{1}$$

where $\boldsymbol{x}_i \in \mathbf{U}_T^o \cup \mathbf{U}_S^o$ and $L = |\mathbf{U}_T^o| + |\mathbf{U}_S^o|$. The output response units $\mathbf{U}_S^o$ are then recovered into the output speech waveform $\mathbf{Y}_S^o$ via a *Unit-to-Speech* (U2S) decoder $d(\cdot, \cdot)$ together with an emotion style embedding $\mathbf{E}_{style}^o$ to realize vivid emotional spoken dialogue controllability (Sec. 3.2).

**LLM.** We adopt the LLaMA-3.1-8B (Dubey et al., 2024) as our base LLM $f(\cdot)$, due to its superior performance among publicly available checkpoints, which is equipped with a tiktoken text tokenizer and a vocabulary size of 128,256, supporting both multilingual textual inputs and outputs.

**Vision encoder and projector.** We utilize InternViT-6B (Chen et al., 2024b) as our visual encoder $v(\cdot)$ with $448 \times 448$ base resolution, and C-Abstractor (Cha et al., 2024) with two ResBlocks (both before and after the pooling layer) and $4\times$ downsample rate as vision projector $p(\cdot)$. To process the high-resolution inputs, the high-resolution image-slicing (Liu et al., 2024a) is used, where visual tokens for one image are concatenation with a low-resolution thumbnail and the origin high-resolution image with separators in each line, allowing a maximum of nine tiles during training.

## 3.2 SPEECH TOKENIZATION

**Speech-to-unit (S2U) tokenizer.** Following Tao et al. (2024), we adopt the SPIRAL (Huang et al., 2022) architecture for the speech encoder $s(\cdot)$ to capture both phonetic and tonal information, which is then discretized by the quantizer $q(\cdot)$ utilizing the finite scalar quantization (FSQ) (Mentzer et al., 2023). The size of the speech codebook is 4,096, while the sample rate is 25 tokens per second. Once discretized, the speech modality can be simply integrated into LLMs by concatenating the text vocabulary and speech codebook.

Our S2U tokenizer provides the following advantages: 1) *Data efficiency*: after pre-training on large-scale unlabeled speech data, it requires only a small amount of speech-text pair data for fine-tuning. 2) *Bilingual*: the speech codebook is shared among different languages (*i.e.*, English and Chinese), sharing unit modeling abilities across languages. Check more training details and comparisons with other speech tokenizers (Zhang et al., 2023b) in Appendix A.1.

**Semantic-acoustic disentanglement.** To seamlessly align speech units with the highly semantic embedding space of LLMs, we opt for decoupling the semantic contents and acoustic styles of input speeches. Specifically, given input speechs $\mathbf{X}_S$, both semantic embedding $\mathbf{E}_{semantic}$ and style embeddings $\mathbf{E}_{style}$ are extracted separately, while only the $\mathbf{E}_{semantic}$ is quantified by $q(\cdot)$ to generate speech units $\mathbf{U}_S$. By changing $\mathbf{E}_{style}$ while maintaining the same $\mathbf{E}_{semantic}$, we can easily control speech styles without disturbing the semantic contents of recovered speeches. Moreover, the disentanglement facilitates modality alignment among speeches and texts, as later shown in Sec. 4.1.

**Unit-to-speech (U2S) detokenizer with style control.** Building on VITS (Kim et al., 2021), our U2S detokenizer adopts a conditional VAE architecture (see Fig. 7). To achieve vivid style controls, we utilize the semantic-style disentanglement of our S2U tokenizer (as discussed above) and adopt a novel style embedding to control the speech styles (*e.g.*, speaker identities, emotions, and pitches). Specifically, the LLM $f(\cdot)$ is trained to generate both the output speech units $\mathbf{U}_S^o$ and a style label. The speech units $\mathbf{U}_S^o$ are converted to unit embeddings $\mathbf{E}_{semantic}^o$, while the style label is utilized to generate a unique style prototype $\mathbf{E}_{style}^o$. Both $\mathbf{E}_{semantic}^o$ and $\mathbf{E}_{style}^o$ are taken as inputs to speech decoder $d(\cdot, \cdot)$ to synthesize the output speech waveform $\mathbf{Y}_S^o$. See Appendix A.2 for more details.

Our U2S detokenizer is pre-trained on LibriTTS (Zen et al., 2019) and AISHELL-1 (Bu et al., 2017) and subsequently fine-tuned on synthetic style-rich speech data. Specifically, due to the scarcity of real-life style-rich data, we utilize TTS tools (Du et al., 2024) to synthesize speech samples diverse in genders, pitches, and emotions. As for style prototypes, Emotion2Vec (Ma et al., 2023) is adopted to select the most representative samples with the highest confidence in conveying the desired style. Our empirical results reveal that even one representative style reference speech has been sufficient to control the speech styles flexibly and precisely.

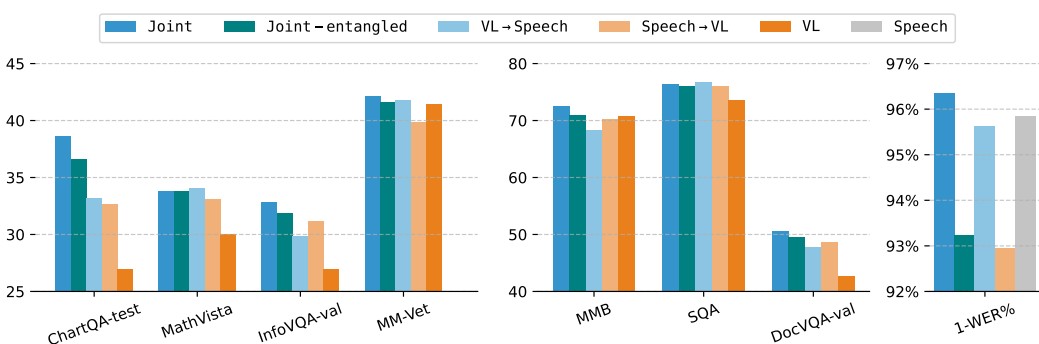

Figure 2: **Comparison between omni-modal alignment paradigms.** 1) `Joint` training achieves consistent improvements over `VL` and `Speech`, suggesting omni-modal alignment can be beneficial across modalities. 2) `Joint` training outperforms both `VL→Speech` and `Speech→VL`, revealing that joint training is more superior and efficient than sequential training. 3) `Joint` is superior to `Joint-entangled`, highlighting the effectiveness of the semantic-acoustic disentanglement.

## 4 OMNI-MODAL ALIGNMENT AND INSTRUCTION TUNING

To achieve the omni-model alignment, it is ideal to utilize large-scale omni-modal image-text-speech data, which, however, is either without reach due to copyright (Nagrani et al., 2022) or limited in the quality (Miech et al., 2019). An alternative is to use existing image-text data with TTS-synthesized speeches, which is not only computationally expensive but also hampers data diversity, as most TTS tools generate speech in similar patterns. Recent works (Chen et al., 2024b; Fu et al., 2024b) choose to integrate the speech modality into a well-structured VLLM via a sequential training manner with **bi-modal** alignment datasets. However, the relationships between different modalities and how to effectively leverage multiple bi-modal alignment datasets remain an open question.

In this work, we explore omni-modal text-centric alignment by utilizing publicly available bi-modal alignment datasets, including both image-text (*e.g.*, captioning) and speech-text (*e.g.*, ASR and TTS) datasets. With the text modality as a bridge, **EMOVA** ultimately becomes a unified system capable of understanding and generating multiple modalities in a coherent and integrated manner. Specifically, in Sec. 4.1, we explore the following three questions:

1. *Does the integration of the speech modality conflict with the vision-language capabilities?*
2. *How to represent speech modality to foster omni-modal alignment?*
3. *Is sequential alignment of multiple modalities optimal?*

Then we introduce the omni-modal instruction tuning pipeline and the overall training paradigm of our **EMOVA** in Sec. 4.2 and Sec. 4.3, respectively.

### 4.1 OMNI-MODAL TEXT-CENTRIC ALIGNMENT

**Settings.** To answer the questions above, we experimentally compare the following omni-modal training paradigms: 1) `VL→Speech` conducts image-text alignment first followed by speech-unit-text alignment using the full speech data and 10% of image-text alignment data to avoid catastrophic forgetting, similar to InternOmni (Chen et al., 2024b) and VITA (Fu et al., 2024b). 2) `Speech→VL` instead performs speech-unit-text alignment first and then aligns images with texts using 10% of the speech unit-text data and the full image-text data. 3) `Joint` aligns both modalities simultaneously. Note that unless otherwise specified, we use the S2U tokenizer introduced in Sec.3.2 to extract speech units for all speech data, which effectively disentangles semantic and acoustic features. `Joint-entangled` derives speech units using HuBERT (Hsu et al., 2021), which does not achieve semantic-acoustic disentanglement effectively with only Kmeans clustering. 4) `VL` and `Speech` only align the vision and speech modalities with texts, respectively, serving as baselines (see Appendix B.1 for more details).

**Evaluation.** For speech capabilities, we directly evaluate the aligned model's performance on the ASR task of LibriSpeech (Panayotov et al., 2015), while for vision-language, we fine-tune the model using a small amount of high-quality visual instruction data (*i.e.*, the 665K SFT data from ShareGPT4V (Chen et al., 2023d)) and evaluate the fine-tuned model on common vision-language

Figure 3: **Demonstration of the omni-modal instruction tuning.** 1) To empower emotional spoken dialogues, **EMOVA** is trained to explicitly select the speech style labels (*e.g.*, emotions and pitches) with output speech units. 2) For the ease of parsing, data components are arranged in `JSON` format.

benchmarks. Check Appendix C for evaluation details. Fig. 2 shows the comparison among different paradigms on vision-language (left and middle) and ASR (right, where we report the $1 - \text{WER}$ value for better readability) benchmarks, from which we can derive the following observations:

**Observation 1: image-text and speech-unit-text data benefit each other.** Contrary to the common assumption that multiple modalities might compete and create conflicts, we find that introducing additional modalities is actually beneficial. As illustrated in Fig. 2, `Joint` consistently outperforms both `VL` and `Speech` across vision-language and speech benchmarks. Moreover, even models aligned sequentially, such as `VL→Speech` and `Speech→VL`, which are typically prone to catastrophic forgetting, demonstrate superior performance on most vision-language tasks. We speculate that the requirement to align multiple modalities with text leads to more robust representations, which in turn generalize better across different downstream tasks. This finding aligns with the results from ImageBind (Girdhar et al., 2023), where the joint alignment of audio and depth modalities with images resulted in improved downstream performance.

**Observation 2: semantic-acoustic disentanglement benefits omni-modal alignment.** We find 1) `Joint` outperforms `Joint-entangled` on vision-language benchmarks, and 2) in the speech tasks, `Joint` maintains significant advantages over its entangled counterpart. This can be attributed to the semantic-acoustic disentanglement which makes speech units more analogous to languages, a domain LLMs are particularly specialized.

**Observation 3: sequential alignment is not optimal.** We notice that `Joint` consistently outperforms its sequential counterparts (*i.e.*, `VL→Speech` and `Speech→VL`) on both vision-language and speech benchmarks, probably due to catastrophic forgetting when integrating a new modality.

In light of these observations, we have chosen to pursue the ultimate alignment strategy that simultaneously aligns image-text and speech-unit-text for **EMOVA**, which offers two important benefits, 1) it fosters mutual enhancement between vision-language and speech capabilities, and 2) it avoids the issue of catastrophic forgetting during sequential alignment of multiple modalities.

## 4.2 OMNI-MODAL INSTRUCTION TUNING

After the omni-modal text-centric alignment, the model learns fundamental vision-language (*e.g.*, captioning) and speech capabilities (*e.g.*, ASR and TTS). However, instruction tuning is essential to better follow complicated user instructions and respond with vivid emotions.

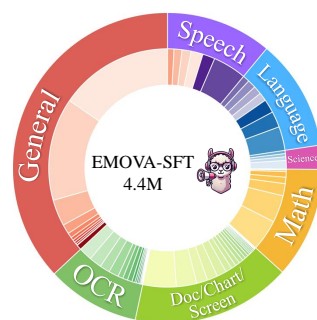

Figure 4: **Overview of the data composition for EMOVA omni-modal instruction tuning.** (Left) Distribution of instruction data across categories, with the outer circle representing overall categories and the inner circle depicting subset distributions. (Right) Quantitative breakdown of data sources.

Table 2: **Detailed configuration for different training stages of EMOVA.** The table illustrates the vision configurations, dataset characteristics, and training hyperparameters.

| | Settings | Stage-1 | Stage-2 | Stage-3 |
|---|---|---|---|---|
| Vision | Resolution | $448 \times \{\{1 \times 2\}, \cdots, \{3 \times 3\}\}$ | $448 \times \{\{1 \times 2\}, \cdots, \{3 \times 3\}\}$ | $448 \times \{\{1 \times 2\}, \cdots, \{3 \times 3\}\}$ |
| | # Tokens | Max $256 \times (1 + 9)$ | Max $256 \times (1 + 9)$ | Max $256 \times (1 + 9)$ |
| Data | Dataset | LCS | EMOVA-Alignment (Fig. 8) | EMOVA-SFT (Fig. 4) |
| | # Samples | 558K | 7.4M | 4.4M |
| Training | Trainable | Projector | Full Model (Half ViT) | Full Model |
| | Batch Size | 256 | 256 | 128 |
| | LR: $p(\cdot)$ | $1 \times 10^{-3}$ | $2 \times 10^{-5}$ | $2 \times 10^{-5}$ |
| | LR: $v(\cdot)$ | - | $2 \times 10^{-5}$ | $2 \times 10^{-6}$ |
| | LR: $f(\cdot)$ | - | $2 \times 10^{-5}$ | $2 \times 10^{-5}$ |
| | Epoch | 1 | 1 | 1 |

**Emotion-enriched instruction data synthesis.** Due to the scarcity of omni-modal instruction data (*i.e.*, dialogues involving images, speeches, and texts simultaneously), we opt for synthesizing omni-modal instruction data from existing text and visual instruction datasets. First, we select instruction data suitable for the vocal expression by filtering out the non-vocal data (*e.g.*, code and mathematical formulas). Second, we clean the selected data to be more vocal by removing text formatting elements (*e.g.*, ** and \n\n). Then, we obtain style labels for the remaining dialog contexts, including genders (male, female), pitches (normal, low, high), and emotions (neutral, happy, sad, angry), resulting in 24 different speech styles. The style labels are obtained by prompting GPT-4o[1] to make reasonable inferences given the dialogue context. Finally, we convert the textual instructions and responses into speeches using the latest TTS tools (*i.e.*, CosyVoice (Du et al., 2024) and Azure AI Speech), and the style labels are used to control the style of the synthesized speech data. To further improve the diversity of the data, each instruction is synthesized by randomly selecting one of the 39 available speakers. Ultimately, we gather 120K speech-text and 110K speech-image data pairs in total. More details can be found in Appendix B.2.

**Data organization and chain of modality.** The omni-modal instruction data can be represented as $D_{\text{omni}} = \{(x_V, u_S, x_T^o, c_{\text{style}}^o, u_S^o)_i\}_{i=1}^N$, where the input consists of the optional queried image $x_V$ and the speech units of the instruction $u_S$, while the output consists of the textual response $x_T^o$, the predicted speech style labels $c_{\text{style}}^o$, and the output speech unit $u_S^o$. Note that we train **EMOVA** to explicitly select styles (*e.g.*, emotions and pitches), which are utilized to determine the corresponding style embedding for the U2S detokenizer (Sec. 3.2). Moreover, since directly generating the speech responses is challenging, we decompose the speech response procedure into three primary steps: 1) recognizing user instructions into texts; 2) generating textual responses based on the recognized instructions; 3) generating the style labels and response speech units based on the textual responses. For ease of parsing during deployment, the target outputs are formatted as JSON, as shown in Fig. 3.

---

[1]https://chatgpt.ust.hk

Table 3: **Comparison on vision-language and speech benchmarks.** 1) **EMOVA** surpasses GPT-4V and Gemini Pro 1.5 on 10 of 14 vision-language benchmarks, while reaching over 95% of GPT-4o performance on nearly all benchmarks. 2) Meanwhile, **EMOVA** achieves state-of-the-art performance on the ASR task, surpassing its speech counterparts by a significant margin.

| Benchmarks | EMOVA 8B | Gemini Pro 1.5 | GPT-4V | GPT-4o | LLaVA-OV-7B | Intern-VL2-8B | Mini-Omni | AnyGPT 7B | VITA 8x7B |
|---|---|---|---|---|---|---|---|---|---|
| MME | 2205 | - | 1927 | 2310 | 1998 | 2215 | - | - | 2097 |
| MMBench | 82.8 | - | 75.0 | 83.4 | 80.8 | 81.7 | - | - | 71.8 |
| SEED-Image | 78.1 | - | 71.6 | 77.1 | - | 75.4 | - | - | - |
| MM-Vet | 55.8 | - | 67.7 | - | 57.5 | 54.3 | - | - | 41.6 |
| RealWorldQA | 64.3 | 68.7 | 61.4 | 75.4 | 66.3 | - | - | - | - |
| TextVQA | 82.0 | 73.5 | 77.4 | - | - | 77.4 | - | - | - |
| ChartQA | 81.8 | 81.3 | 78.5 | 85.7 | 80.0 | 83.3 | - | - | - |
| DocVQA | 90.4 | 86.5 | 88.4 | 92.8 | 87.5 | 91.6 | - | - | - |
| InfoVQA | 64.4 | 72.7 | - | - | 68.8 | 74.8 | - | - | - |
| OCRBench | 824 | - | 656 | 736 | - | 794 | - | - | 678 |
| MathVista | 61.1 | 52.1 | 49.9 | 63.8 | 63.2 | 58.3 | - | - | 44.9 |
| Mathverse | 27.8 | - | 33.6 | - | 26.2 | - | - | - | - |
| ScienceQA-Img | 94.0 | - | 75.7 | - | 96.0 | 97.1 | - | - | - |
| AI2D | 82.8 | 80.3 | 78.2 | 84.6 | - | 83.8 | - | - | 73.1 |
| Librispeech (WER↓) | 4.0 | - | - | - | - | - | 4.5 | 8.5 | 8.1 |

## 4.3 OVERALL TRAINING PARADIGM

Inspired by Chen et al. (2023d), a three-stage training paradigm is adopted for **EMOVA**,

- **Stage-1: Vision-language pre-alignment.** The purpose is to align visual features into the embedding space of LLMs. Only the vision projector $p(\cdot)$ is trained.
- **Stage-2: Omni-modal text-centric alignment.** This stage jointly performs the vision-language and speech-language alignment simultaneously. We train the LLM $f(\cdot)$, vision projector $p(\cdot)$, and the deeper half of vision encoder $v(\cdot)$ layers.
- **Stage-3: Omni-modal instruction tuning.** To empower **EMOVA** to respond accurately to omni-modal instructions, we organize different datasets with various types of instructions to enforce **EMOVA** to learn generalization across tasks, as detailed in Sec. 5.1.

## 5 EXPERIMENTS

### 5.1 TRAINING CONFIGURATION

**Stage-1.** In this stage, we only train the parameters of the vision projector $p(\cdot)$ for vision-language pre-alignment with the LCS-558K dataset (Liu et al., 2024b), with the high-resolution image-slicing strategy (Liu et al., 2024a) adopted.

**Stage-2.** We assemble a unified dataset with 7.4M samples for both the image-text and speech-text alignment, as summarized in Fig. 8. Specifically, we utilize pre-training datasets from ShareGPT4V (Chen et al., 2023d), ALLaVA (Chen et al., 2024a) (both the original English version and the Chinese version translated on our own), and ShareGPT-4o (Cui et al., 2023) for general perception, while for the OCR capabilities, we leverage SynthDog (Kim et al., 2022), MMC-Alignment (Liu et al., 2023a), K12 Printing, and UReader Text Reading subset (Ye et al., 2023). Moreover, we use the 2,000 hours of ASR and TTS data from LibriSpeech (Panayotov et al., 2015) and AISHELL-2 (Du et al., 2018) for speech-text alignment, and to preserve the language capabilities of LLMs, we further incorporate the text-only data from Magpie Pro (Xu et al., 2024).

**Stage-3.** We collect the EMOVA-SFT dataset consisting of 4.4M multi-task omni-modal samples (see Fig. 4). We start by gathering high-quality open-sourced visual instruction datasets, including ShareGPT4V (Chen et al., 2023d), InternVL (Chen et al., 2024b), Meteor (Lee et al., 2024), Idefics-2 (Laurençon et al., 2024), Cambrian (Tong et al., 2024), and LLaVA-Onevision (Li et al., 2024a), followed by quality checking, re-formatting all data samples to a unified template, and removing the duplicates. For speech, we include the training split of **EMOVA** omni-model instruction data (*c.f.*, Sec. 4.2), with 10% of speech alignment datasets to maintain ASR and TTS performance. We adopt 128 Ascend 910 Training NPUs in parallel for model training (check more details in Table 2).

Table 4: **The performance of EMOVA on Speech Dialogue tasks.** By default, we evaluate on the corresponding test set of the evaluated datasets.

| Datasets | End-to-end↑ | Text response | | Style Categorization | | Recognition/Synthesis | |
|---|---|---|---|---|---|---|---|
| | | Unit In | Text In | Emotion | Pitch | WER/CER↓ | TTS-WER/CER↓ |
| Speech-Image-EN | 7.21 | 7.43 | 7.75 | 89.01 | 98.30 | 2.45 | 4.60 |
| Speech-Text-EN | 6.90 | 7.02 | 8.27 | 87.26 | 90.50 | 6.80 | 6.75 |
| Speech-Image-ZH | 5.87 | 6.51 | 6.44 | 92.60 | 96.60 | 1.85 | 13.95 |
| Speech-Text-ZH | 4.70 | 4.88 | 6.00 | 87.19 | 94.40 | 12.60 | 9.70 |

User 🔊 Write a poem for this image.    EMOVA 🦙 🔊 *Emotion: Sad; Pitch: Low.*

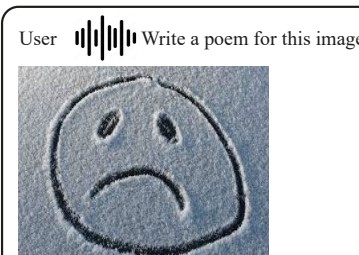

In winter's chill, a sad face drawn, A frown of snow, a somber tone. The world is quiet, still, and cold, A canvas painted by the frost's cold. The eyes, two dots, the mouth, a line, A simple face, a symbol of decline. The snowflakes dance, they twirl and play, A fleeting moment, soon to fade away. But in this brief sketch, a story's told, Of winter's mood, of solitude, of cold. So let this face remind us all, In winter's chill, we find our call. To feel the frost, to feelthe snow, To find the beauty in its woe.

Figure 5: **EMOVA** engages in emotional spoken dialogue expressing sadness.

## 5.2 COMPARISON TO THE SOTA MODELS

Experimental results are provided in Table 3. We compare a wide range of state-of-the-art VLLMs, including proprietary ones like Gemini Pro 1.5 (Reid et al., 2024), GPT-4V (OpenAI, 2023), GPT-4o (OpenAI, 2024), and open-sourced ones like LLaVA-OneVision-7B (Li et al., 2024a), InternVL2-8B (Chen et al., 2024b), together with the Speech LLM Mini-Omni (Xie & Wu, 2024) and omni-modal LLM AnyGPT (Zhan et al., 2024) and VITA-8x7B (Fu et al., 2024b).

**Comparison with SOTA VLLMs.**  As an omni-modal model, **EMOVA** achieves comparable performance with state-of-the-art open-sourced VLLMs across multiple benchmarks. Specifically, our **EMOVA** outperforms both InternVL2 and LLaVA-OV on MMBench, SEED-Image, TextVQA, and OCRBench consistently, while demonstrating exceptional proficiency in solving math problems requiring precise visual content interpretation. **EMOVA** achieves a 2.8% improvement on MathVista compared with InternVL2 and 1.6% higher score on Mathverse compared to LLaVA-OneVision.

Furthermore, **EMOVA** shows competitive performance even compared with the proprietary models. **EMOVA** exceeds both GPT-4V and Gemini Pro 1.5 significantly on **10 out of 14** benchmarks, while for GPT-4o, **EMOVA** outperforms on both SEEDBench-Image and OCRBench, reaching over 95% of GPT-4o's performance on ALL evaluated benchmarks except RealWorldQA.

**Comparison with SOTA omni-modal LLMs.**  Compared with VITA-8x7B, **EMOVA** shows substantial improvement on visual-language benchmarks. Specifically, **EMOVA** is 112 points higher than VITA on MME, and surpasses VITA by 21.5% on OCRBench, underscoring the effectiveness of our approach and the potential to push boundaries of omni-modal abilities. What's more, **EMOVA** significantly outperforms the most recent omni-modal model VITA, even surpassing its SLLM counterpart Mini-Omni, showing the effectiveness of the semantic-acoustic disentanglement and omni-modal mutual benefits. Qualitative results are shown in Fig. 5 and Appendix G. We also report TTS results in Table 6. For the first time, our **EMOVA** obtains state-of-the-art performance on both the vision-language and speech benchmarks simultaneously.

## 5.3 EVALUATION OF EMOTION-RICH SPOKEN DIALOGUE

In this section, we evaluate the end-to-end spoken dialogue capabilities of **EMOVA**. As discussed in Sec. 4.2, the model takes an input image $x_V$ and user instructions in the form of speech units $u_S$, and outputs *text responses*, *style labels*, and *corresponding speech units*. To ensure comprehensive evaluation, we propose the following evaluation metrics (see Appendix D for more details):

1. **End-to-end spoken dialogue score** assesses the model's dialogue performance based on the generated speeches, with a score ranging from 0 to 10, reporting the average.
2. **Unit-input-text-output score** focuses on the quality of the text responses of LLM when the inputs are speech units, bypassing errors from speech synthesis.

3. **Text-input-text-output score** inputs the ground-truth text of the user instruction and evaluate the model's text output. This helps disentangle the impact of speech recognition errors and eliminates the influence of the `JSON` format.

4. **ASR and TTS** evaluate how accurately **EMOVA** recognizes speech units and how effectively it generates speech units from text.

5. **Style label classification accuracy** evaluates the accuracy of the model in selecting the appropriate speech style labels (Sec. 3.2).

6. **Style controllablity** assesses the style controllability of U2S detokenizer with the given conditional style labels via the confusion matrix between generated and recognized styles.

Due to the lack of emotionally rich spoken dialogue evaluation datasets, we split a test set from our synthesized omni-modal instruction-tuning data (see Sec. 4.1). GPT-4o are used for automated evaluation. Details are provided in Appendix D.

**Results** are shown in Table 4. As can be seen,

**(i)** By comparing the *end-to-end dialogue score* with the *unit-input-text-output score*, we notice that the two scores are closely aligned, with a maximum gap of only 0.22, except for Speech-Image-ZH. The TTS-WER/CER is generally low, revealing that **EMOVA** can synthesize accurate speech based on text responses. However, the Speech-Image-ZH is an outlier, which we attribute to its complexity. It includes tasks such as generating poetries and answering riddles, resulting in more intricate responses. When these answers are converted to speeches and then transcribed back to texts, multiple variations often arise, leading to discrepancies from the original responses.

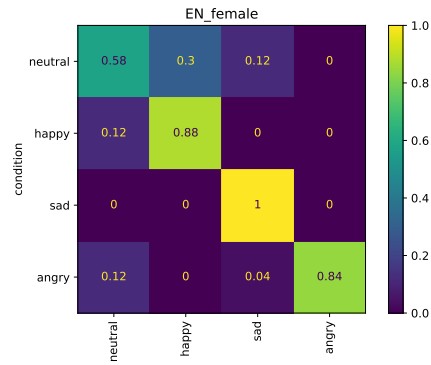

Figure 6: **Confusion matrix between the generated and recognized emotions.**

**(ii)** Comparing the *unit-input-text-output* score with the *text-input-text-output* score, we observe that their differences correlate with the ASR results of speech instructions. Specifically, for Speech-Text-EN and Speech-Text-ZH, which involve more complex instructions, **EMOVA** reports inferior ASR performance (6.8 and 12.6, respectively) compared to other datasets (2.45 and 1.85). Consequently, when we replace speech instructions with ground-truth transcriptions, **EMOVA** shows significant improvements from *unit-input* to *text input* score. On the contrary, for datasets with accurate ASR performance, the results are quite similar, suggesting **EMOVA** retains robust dialogue capabilities when using the `JSON` format.

**(iii)** Examining the *classification accuracy of style labels*, we find that **EMOVA** performs satisfactorily in classifying emotions and pitch during speech conversations, achieving an accuracy of over 80%. The confusion matrix comparing the conditional and recognized emotion labels is shown in Fig. 6. The results indicate that the four emotions are recognized with high probabilities, with three achieving over 80% accuracy. This demonstrates that our U2S detokenizer effectively controls common emotions, endowing the synthesized speech with vivid emotional expression.

## 6 CONCLUSION

This work presents **EMOVA**, an innovative end-to-end omni-modal large language model that effectively aligns vision, speech, and text simultaneously. We employ a continuous vision encoder to capture fine-grained visual details, while a discrete, semantic-acoustic disentangled speech tokenizer and detokenizer enable end-to-end speech understanding and generation. A lightweight style module further supports spoken dialogue with vivid emotions. By using text as a bridge, we demonstrate that omni-modal alignment is achievable without relying on scarce omni-modal image-text-speech data, which not only enhances both vision-language and speech capabilities but also surpasses its bi-modal counterparts through joint optimization. For the first time, **EMOVA** achieves state-of-the-art performance on both vision-language and speech benchmarks, setting a novel standard for the omni-modal models for versatile and expressive omni-modal interactions.

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

APPENDIX

# A  MORE ON SPEECH TOKENIZER

## A.1  SPEECH-TO-UNIT (S2U) TOKENIZER

**Overview.**  To process the speech input $\mathbf{X}_S$, our S2U tokenizer consists of a speech encoder $s(\cdot)$ with a quantization module $q(\cdot)$. First, the speech input is passed through $s(\cdot)$, producing a continuous latent representation $\mathbf{E}_S = s(\mathbf{X}_S)$. Then, the quantization module $q(\cdot)$ converts $\mathbf{E}_s$ into discrete units $\mathbf{U}_S = q(\mathbf{E}_s)$. The final output is an ID sequence $\mathbf{U}_S = [u_1, u_2, \cdots]$, where each $u_i$ corresponds to a unique speech unit.

After this S2U extraction process, the speech is represented by quantized IDs instead of quantized embeddings. For example, a speech signal is represented as an ID sequence like $[782, 463, 550, \cdots]$, which can be treated as a special form of text. As a result, to integrate speech tokens into LLM $f(\cdot)$, we only need to expand the LLM's original vocabulary $V_T$ by adding a set of speech unit tokens $V_S$, similar to Zhang et al. (2023a). The expanded vocabulary is thus the union $V = V_T \cup V_S$. In this work, the same codebook is shared across multiple languages, such as English and Chinese, enabling the unit modeling abilities to be shared across languages. This design enables simply integration of the speech modality into the LLM with minimal data requirements (see experiments in Sec. 4.1).

**Training of S2U.**  The training of the S2U encoder involves three primary components: the speech encoder, the quantization module, and the phone decoder. First, the speech encoder is trained with a large amount of unlabeled speech with contrastive loss in a self-supervised learning manner (Tao et al., 2024). The dataset utilized is the 10000-hour English speeches from GigaSpeech (Chen et al., 2021a) and the 10000-hour Chinese speeches from WenetSpeech (Zhang et al., 2022), both of which encode large variations in speakers, styles, and acoustic environments. Subsequently, the speech encoder, along with the quantization module and phone decoder, are optimized on a small amount of speech-text pair data, which is derived from the train-clean-100 subset of LibriSpeech (Panayotov et al., 2015) in English and AISHELL-1 (Bu et al., 2017) in Chinese. The phone label is obtained using the phone set in Charsiu (Zhu et al., 2022). During training, the speech encoder encodes input speeches into continuous latent representations that are rich in both phonetic and tonal information. Then, the quantization module is used to convert the continuous outputs from the speech encoder into discrete units. Finally, the phone decoder converts the quantized embeddings into a sequence of non-tonal/tonal phones, ensuring that the speech units capture necessary information related to semantic contents in both non-tonal and tonal languages. After training, only the speech encoder and the quantization module are used in **EMOVA**.

**Comparisons with SpeechTokenizer in AnyGPT.**  Our S2U tokenizer differs from the Speech-Tokenizer (Zhang et al., 2023b) used in AnyGPT (Zhan et al., 2024), in the following aspects:

(1) SpeechTokenizer encodes both semantic contents and acoustic details of speeches, while our S2U tokenizer focuses solely on semantic contents. This design reduces the LLMs' burden of disentangling different aspects of speech information, facilitating the modality alignment between speech and text modalities during LLM training.

(2) Compared with SpeechTokenizer, our S2U tokenizer offers a more concise representation and helps to simplify and accelerate the generation of **EMOVA**. SpeechTokenizer employs tokens from eight RVQ layers with a 50Hz frame rate to represent speech, thus a 10-second speech corresponds to $500 \times 8 = 4000$ tokens. However, we reduce the frame rate from 50Hz to 25Hz and utilize only one token to represent each frame, and thus, a 10-second speech can be represented by only 250 tokens. Moreover, AnyGPT requires a two-stage generation process, involving autoregressive (AR) semantic token generation followed by the non-autoregressive (NAR) acoustic token generation. Instead, we only need to generate speech units capturing semantic contents in a fully AR manner.

(3) SpeechTokenizer lacks an explicit structure design to deal with tonal languages like Chinese, therefore, the processing ability in Chinese is not demonstrated in either SpeechTokenizer or AnyGPT. In contrast, our S2U tokenizer incorporates training constraints to better capture tone variation in phone, making it effective for both the non-tonal and tonal languages. This further en-

hances **EMOVA**'s multilingual speech processing capabilities, enabling it to effectively handle both English and Chinese.

In summary, our S2U tokenizer improves the compactness and generality of speech representation, facilitates LLM training, and enhances its multilingual speech ability. Experimental results show that our model significantly outperforms AnyGPT in ASR tasks, as shown in Table 6.

### A.2 UNIT-TO-SPEECH (U2S) DETOKENIZER WITH STYLE CONTROL

**Overview.** The LLM, along with the vision encoder and speech tokenizer, is trained end-to-end to generate responses in the form of the speech units, given the input images and speeches. Specifically, the output speech units can be obtained via $\mathbf{U}_S^o = f(\mathbf{U}_T, \mathbf{U}_S, \mathbf{H}_V)$, followed by a U2S detokenizer to convert the discrete speech units $\mathbf{U}_S^o$ into the final output speech waveforms.

The proposed U2S detokenizer involves three core modules: the speech unit encoder $e(\cdot)$, the speech style encoder $g(\cdot)$, and the speech decoder $d(\cdot, \cdot)$. First, the speech unit encoder converts the speech units $\mathbf{U}_S^o$ into unit embeddings $\mathbf{E}_{unit}^o$. Meanwhile, the style encoder $g(\cdot)$, adopting the structure of Meta-StyleSpeech (Min et al., 2021), is utilized to extract a style embedding $\mathbf{E}_{style}^o$ from the chosen reference speech. Lastly, the speech decoder $d(\cdot, \cdot)$ reconstructs the speech waveform $\mathbf{Y}_S^o$ from the unit embedding $\mathbf{E}_{unit}^o$ and style embedding $\mathbf{E}_{style}^o$.

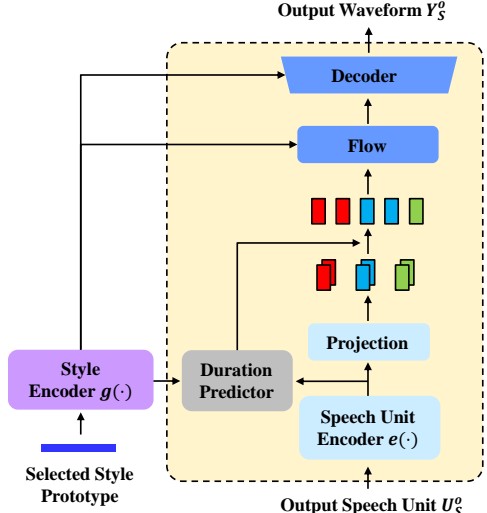

Figure 7: **U2S detokenizer with style control.**

**Training of U2S.** Training a U2S detokenizer with emotion controls is challenging considering the lack of labeled emotional speech data since most open-source speech data is predominantly neutral in emotion or lacks emotion labels. Due to the limited availability of emotion-rich data, we utilize TTS tools (Du et al., 2024) to generate a small set of style-rich speech samples diverse in speaker identities, genders, emotions, and pitches. Our U2S detokenizer is first pre-trained on LibriTTS (Zen et al., 2019) and AISHELL-1 (Bu et al., 2017) to acquire fundamental speech synthesis capabilities, and subsequently, the synthesized style-rich speech data is utilized to fine-tune the U2S detokenizer, enhancing its controllability over diverse speech styles.

**Style Prototypes.** To better facilitate controls of genders, emotions, and pitches, inspired by Min et al. (2021) that a small number of style reference speeches can effectively transfer the target styles, we adopt a "store-for-usage" manner, *i.e.*, we construct a style prototype codebook in advance for speech style assignation. Specifically, we synthesize $K$ reference candidates with external TTS tools for each possible combination of the following styles: two genders (`male, female`), seven emotions (`neutral, happy, sad, angry`), and three pitches (`normal, high, low`), leading to 24 unique styles and $24 \times K$ candidates. Empirically we find that genders and pitches are easy to control using any of the candidate references, while the emotion intensity varies across speeches. To tackle this, we adopt Emotion2Vec (Ma et al., 2023), a powerful speech emotion recognition (SER) tool, to measure the emotion intensity of each candidate reference, and rank them in terms of the confidence of the desired emotion. We select the Top-1 candidate reference in each combination style to be the prototype of this condition. Finally, the most representative 24 reference speeches are selected from the $24 \times K$ candidates.

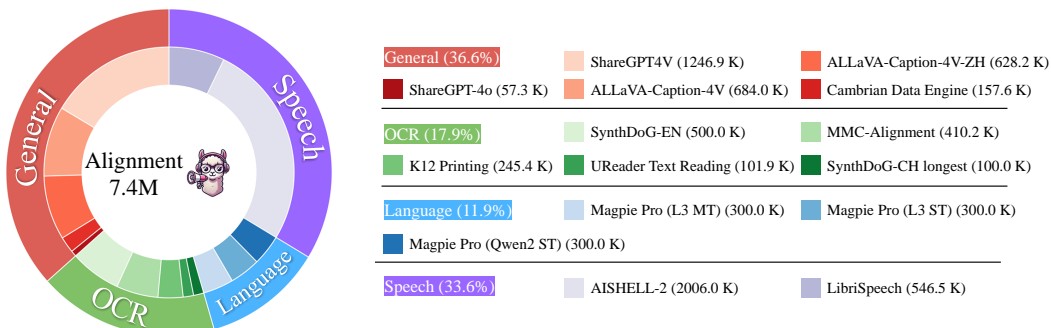

Figure 8: **Overview of EMOVA omni-modal alignment data composition.**

# B  MORE ON OMNI-MODALITY

## B.1  OMNI-MODAL TEXT-CENTRIC ALIGNMENT

**Modality alignment data** is summarized in Fig. 8.

**Experiments on Omni-modal Alignment Paradigms.** The training configuration adopted in Sec. 4.1 is mostly identical to Table except that we use a unique resolution of 448 for all stages and replace EMOVA-SFT in Stage-3 with ShareGPT4V (Chen et al., 2023d) for efficiency.

Given the space constraints, the evaluation benchmarks in Fig. 2 represent selected benchmarks from each category in Table 3. Specifically, for general image perception and understanding, we choose MMBench and MM-Vet; for mathematical problem solving, we adopt MathVista (testmini); for science understanding, we select ScienceQA-Img; and for automatic speech recognition (ASR), we utilize the test-clean split of the LibriSpeech dataset.

## B.2  OMNI-MODAL INSTRUCTION DATA SYNTHESIS

**Dataset construction.** To obtain emotion and pitch labels, we leverage GPT-4o using the prompt in Fig. 22. Table 5 shows the distribution of speech styles of our speech instruction dataset.

**Detailed data organization.** As discussed in Sec. 4.2, the omni-modal instruction data is formulated as $D_{\text{omni}} = \{(x_V, u_S, x_T^o, c_{\text{style}}^o, u_S^o)_i\}_{i=1}^N$. In details, the textual outputs $x_T^o = (x_T^{o^1}, x_T^{o^1})$ contain the transcribed textual instructions $x_T^{o^1}$ and the textual responses $x_T^{o^2}$. The styles labels $c_{\text{style}}^o = (c_{\text{emo}}^o, c_p^o)$ include the emotion and pitch labels, respectively.

**Mathematical formulation of chain of modality.** Based on the notations above, the sequential chain of modality approach can be mathematically formulated by decomposing the conditional likelihood of the desired outputs $(x_T^{o^1}, x_T^{o^1}, c_{\text{emo}}^o, c_p^o, u_S^o)$ given the inputs $(x_V, u_S)$. Specifically, let $z_1 = x_T^{o^1}, z_2 = x_T^{o^1}, z_3 = c_{\text{emo}}^o, z_4 = c_p^o,$ and, $z_5 = u_S^o$, the decomposition is expressed as:

$$\mathbb{P}(x_T^{o^1}, x_T^{o^1}, c_{\text{emo}}^o, c_p^o, u_S^o \mid x_V, u_S) = \prod_{i=1}^{5} \mathbb{P}(z_i \mid z_{1:i-1}, x_V, u_S). \tag{2}$$

# C  MORE ON BENCHMARK EVALUATION

To thoroughly evaluate our model's vision-language abilities, 14 benchmarks covering four different aspects of real-life scenarios are utilized for a comprehensive assessment across multiple domains. Moreover, Automatic Speech Recognition (ASR) and Text-to-speech (TTS) are adopted to evaluate speech-language abilities.

Table 5: **Statistics of the EMOVA speech instruction tuning datasets.**

| Dataset | Source | # Examples | Emotions | | | | Pitches | | |
|---|---|---|---|---|---|---|---|---|---|
| | | | Neutral | Happy | Sad | Angry | Normal | Low | High |
| Speech-Image -EN(train) | ALLAVA | 71,474 | 58,506 | 12,412 | 516 | 40 | 70,962 | 392 | 120 |
| Speech-Image -EN(test) | ALLAVA | 1,056 | 434 | 274 | 300 | 48 | 44 | 176 | 16 |
| Speech-Image -ZH(train) | ALLAVA (ZH) | 38,260 | 29,893 | 7,680 | 607 | 80 | 36,363 | 624 | 1,273 |
| Speech-Image -ZH(test) | ALLAVA (ZH) | 616 | 96 | 193 | 190 | 137 | 381 | 177 | 58 |
| Speech-Text -EN(train) | ShareGPT | 73,658 | 42,334 | 20,946 | 4,674 | 5,704 | 60,352 | 5,518 | 7,788 |
| Speech-Text -EN(test) | ShareGPT | 1,400 | 200 | 400 | 400 | 400 | 582 | 422 | 422 |
| Speech-Text -ZH(train) | In-house | 47,936 | 29,769 | 16,405 | 1,446 | 316 | 42,356 | 4,379 | 4,379 |
| Speech-Text -ZH(test) | In-house | 686 | 96 | 196 | 198 | 196 | 458 | 134 | 92 |

**Document/chart understanding and OCR abilities.** Benchmarks including the TextVQA (Singh et al., 2019), ChartQA (Masry et al., 2022), DocVQA (Mathew et al., 2021), InfoVQA (Mathew et al., 2022), and OCRBench (Liu et al., 2023c), focus on recognition and understanding of structured data (*e.g.*, charts, documents, and characters), challenging the model to extract, comprehend, and reason with structural and textual data. We adopt the corresponding test set for evaluation.

**General image perception and understanding.** MME (Fu et al., 2024a), MMBench (Liu et al., 2023b), SEED-Bench-Image (Li et al., 2023a), MM-Vet (Yu et al., 2024), and RealWorldQA (xAI, 2024) evaluate models on perception and reasoning among general vision domains, providing a comprehensive measurement of models' real-world generalization abilities.

**Mathematical problem solving.** MathVista (testmini) (Lu et al., 2024) and MathVerse (vision-mini) (Zhang et al., 2024) evaluate the model's ability to deal with diverse mathematical problems, including both arithmetic and complex reasoning questions across different levels of complexity.

**Science understanding.** ScienceQA-Img (Lu et al., 2022) and AI2D (Kembhavi et al., 2016) are utilized to assess the model's ability to deal with scientific questions and diagrams, which requires reasoning across various subjects and interpreting structured visual content.

**Automatic speech recognition (ASR).** We utilize the test-clean set of LibriSpeech (Panayotov et al., 2015) for English, reporting the Word Error Rate (WER) as the evaluation metric. For Chinese, evaluation is conducted on the test set of AISHELL-2 (Du et al., 2018), using the Character Error Rate (CER). Both WER and CER assess ASR performance, calculated by comparing the recognized texts with the ground-truth transcripts.

**Text-to-speech (TTS).** To evaluate the TTS abilities, we first prompt **EMOVA** to generate speech units, which are then converted to speech waveforms by the U2S detokenizer. Using the synthesized speech as input, we conduct ASR with Whisper-large-v3 and Paraformer-zh for English and Chinese, respectively, to obtain transcribed texts. We then compute the WER and CER between the ground truth texts and the transcribed texts as metrics for TTS. The resulting metrics are denoted as TTS-WER and TTS-CER for English and Chines.

# D MORE ON EVALUATION OF SPEECH-LANGUAGE CAPABILITIES

## D.1 CALCULATION OF EVALUATION METRICS

**End-to-end spoken dialogue score.** We prompt GPT-4o with the original question $x_T^{o^1}$, the ground-truth text answer $x_T^{o^2}$ and the transcribed text from the generated speech, to obtain a score ranging from 0 to 10 and report an average of them. The prompt can be found in Fig. 24.

Table 6: **Comparison on the ASR and TTS benchmarks.**

| Models | Librispeech (EN) | | AISHELL-2 (ZH) | |
| --- | --- | --- | --- | --- |
| | WER↓ | TTS-WER↓ | CER↓ | TTS-CER↓ |
| Mini-Omni (Xie & Wu, 2024) | 4.5 | - | - | - |
| AnyGPT (Zhan et al., 2024) | 8.5 | - | - | - |
| VITA (Fu et al., 2024b) | 8.1 | - | - | - |
| **EMOVA (ours)** | **4.0** | 3.4 | 10.3 | 7.9 |

**Unit-input-text-output score.** Similar to end-to-end spoken dialogue score, but we use the predicted text response $\tilde{x}_T^{o2}$ as answer instead of the transcribed text from the generated speech, to obtain a score ranging from 0 to 10 and report an average of them. See the prompt in Fig. 23.

**Text-input-text-output score.** The prompt can be found in Fig. 23.

**Style label classification accuracy.** We use GPT-4o to decide whether the style predictions $\tilde{c}_{\text{emo}}^{o}, \tilde{c}_{\text{p}}^{o}$ are correct given the transcribed instruction $\tilde{x}_T^{o1}$ and the predicted text response $\tilde{x}_T^{o2}$. The prompt can be found in Fig. 25.

**Emotion controllablity** of our U2S detokenizer is assessed by providing texts to LLM to generate corresponding units (*i.e.*, TTS), which, along with the given conditional emotion labels, are then fed into our U2S detokenizer to synthesize speech. We choose female voice due to its large variation of styles. We evaluate on 4 commonly-seen emotion, *i.e.*,"neutral", "happy", "sad", and "angry". We synthesize 200 speech utterances for testing, with 50 utterances per emotion. The output speeches are analyzed by a Speech Emotion Recognition (SER) model named Emotion2Vec (Ma et al., 2023), which identifies the emotion with the greatest likelihood among these four emotion.

### D.2 COMPARISON WITH OTHER OMNI MODELS

Experimental results of ASR and TTS are reported in Table 3 and 6. **EMOVA** achieves significant improvements over other omni-modal models (*i.e.*, AnyGPT and VITA), even surpassing its SLLM counterpart Mini-Omni (Xie & Wu, 2024), demonstrating the effectiveness of semantic-acoustic disentanglement and omni-modal mutual benefits. For the first time, our **EMOVA** obtains state-of-the-art performance on both the vision-language and speech benchmarks simultaneously.

## E MORE ON VISION-LANGUAGE

Table 7: **Ablation on the ViT configurations and templates for vision-language alignment.**

| ViT | ViT LR | Template | MME | MMBench | SEED-Image | TextVQA | ChartQA | DocVQA | InfoVQA | OCRBench | ScienceQA-Img | AI2D |
| --- | --- | --- | --- | --- | --- | --- | --- | --- | --- | --- | --- | --- |
| Full | $2\times10^{-6}$ | QA | **1928** | 68.8 | 72.5 | **64.3** | 29.9 | 45.2 | 28.7 | **495** | 76.3 | 61.8 |
| Half | $2\times10^{-6}$ | QA | 1838 | **71.3** | 72.8 | 63.3 | **31.4** | **46.0** | 28.5 | 489 | 76.1 | **63.7** |
| Frozen | $2\times10^{-6}$ | QA | 1887 | 68.8 | 72.2 | 61.3 | 30.2 | 44.7 | 28.0 | 478 | 75.9 | 62.8 |
| Half | $2\times10^{-5}$ | QA | 1833 | 68.3 | **73.1** | 62.2 | 27.8 | 42.4 | 27.3 | 461 | 74.7 | 63.4 |
| Half | $2\times10^{-6}$ | Plain | 1909 | 70.1 | 72.0 | 61.5 | 24.5 | 38.9 | **30.1** | 410 | **77.0** | 63.6 |

This section analyzes the pre-training configurations of the vision encoder and the prompt template during Stage 2, as shown in Table 7. Our final selection is the colored setting. We find that training the ViT model with half of the deeper parameters (Chen et al., 2023d) with a learning rate of $2\times10^{-6}$ (Liu et al., 2024a) yields the best performance. Furthermore, we compare the QA template with the plain template (Liu et al., 2024b) and find that the QA template is superior for pre-training.

## F LIMITATIONS

**Duplex modeling.** In the current version, **EMOVA** can only process either visual/speech/text inputs or produce speech/text outputs at the same time. For a communication experience that mirrors human interaction, handling inputs and outputs simultaneously is crucial. Recent works like VITA

(Fu et al., 2024b) and LSLM (Ma et al., 2024) have begun to explore duplex modeling. VITA focuses on recognizing speech in noisy environments during the generation process to facilitate timely responses. LSLM, on the other hand, attempts to halt speech production when it detects a command or voice. Recently, a ground-breaking work named Moshi (Défossez et al., 2024) develops a model supporting fully duplex modeling. The adeptness at simultaneously managing the information streams from both the user and the assistant allows Moshi to converse with human beings in real-time scenarios.

However, incorporating emotion into this duplex modeling presents additional challenges. Emotional states can fluctuate throughout a conversation, and how to generate appropriate responses given the user's previous and current emotional cues has not been thoroughly investigated. We will dive into this topic in the future work.

**Direct unit-to-unit generation.** Although speech units have served as speech representation, they are predominantly adopted in conjunction with text-based assistance (Zhang et al., 2023a). However, the direct generation from unit to unit without text assistance is an area that has not been extensively explored. In Lee et al. (2021), speeches from the source language are directly translated into speech units of the target language for speech-to-speech translation. Similarly, Nguyen et al. (2023) builds a language model directly on speech units, enabling spoken dialogue generation from raw audio. Both works develop models in speech-only data.

In the current version of **EMOVA**, the text modality is integrated into the speech generation process to transfer textual knowledge to the speech modality, thereby enhancing the correctness of speech responses. In the future, we will strengthen the model's direct unit-to-unit generation capabilities to boost the speed of speech generation and augment the model's comprehension of speech modality.

**Vision configurations.** Currently, we only utilize a single vision encoder pre-trained via a vision-language manner, while recent works have shown effectiveness by combining vision encoders pretrained by different manners (*e.g.*, self-supervised pre-training (Liu et al., 2022)) and architectures (*e.g.*, MoE (Zhili et al., 2023; Liu et al., 2024c)). We prioritize visual understanding in this work, while the incorporation of (controllable) visual generation (Chen et al., 2023c; Gao et al., 2023; Li et al., 2023b; Wang et al., 2024; Liu et al., 2023d; Gao et al., 2024) is also appealing to better empower **EMOVA** in real-life applications (Li et al., 2024c; 2022; Han et al., 2021).

## G QUALITATIVE RESULTS

**Advanced Vision-language Abilities.** **EMOVA** exhibit advanced vision-language abilities in humor understanding (Fig. 9), numerical calculations (Fig. 10), coding (Fig. 11), geometry problem solving (Fig. 15).

**(Omni-modal) Emotional Spoken Dialogue.** **EMOVA** can engage in omni-modal emotional spoken dialogue (Figs. 5, 12, 13, 17). For demonstration purposes, we present only the speech transcriptions[2] of the instruction and EMOVA's response. The text highlighted in blue indicates the style labels predicted by EMOVA, reflecting the emotion and pitch of the generated speech.

**Omni-modal Spoken Dialogue with Structural Data Understanding.** **EMOVA** can perform structural data understanding in spoken dialogue. The data types include PDF files (Fig. 14), movie posters (Figs. 16 and 18), personal résumés (Fig. 19), charts (Fig. 20), and websites (Fig. 21). Remarkably, EMOVA is not specifically trained on spoken dialogue involving these data types. This suggests that our chain of modality data introduced in Sec. 4.2 effectively activates both vision-language abilities and speech understanding acquired during the text-centric alignment stage.

---

[2]For speech files, please refer to our project page.

**Advanced Vision-language Abilities (Humor Sense)**

**Me:** I'm going to sleep early tonight
**Also me at 2am:**

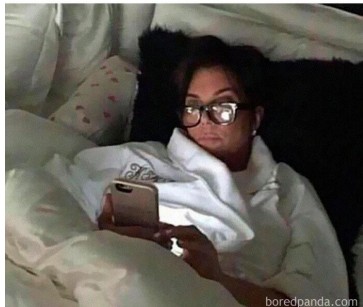

| User | Why is this image funny? |
|---|---|
| EMOVA | The humor in this image comes from the juxtaposition of the woman's statement about going to sleep early and the fact that she is still awake at 2am, as indicated by the text. The woman's expression and the use of a smartphone in bed suggest that she might be scrolling through social media or engaging in some other activity that is keeping her awake. This contrast between her intention to sleep early and her current state of being awake creates a comedic effect, as it highlights a common human experience where plans don't always go as intended. The meme format, which typically presents a situation with an ironic or humorous twist, amplifies this effect by presenting the woman's situation in a relatable and exaggerated way. |

Figure 9: **EMOVA** exhibits **advanced vision-language abilities in humor sense**.

**Advanced Vision-language Abilities (Calculation)**

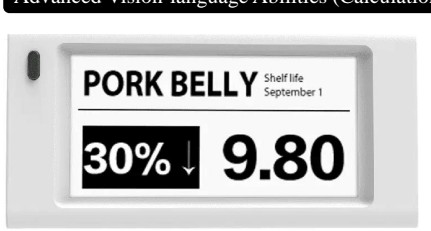

| User | First please perform reasoning, and think step by step to provide best answer to the following question: What is the original price for pork belly before discount? |
|---|---|
| EMOVA | To find the original price of pork belly before the discount, we need to use the information provided in the image. The image shows that the price after a 30% discount is $9.80. This price represents 70% of the original price, since 100% - 30% = 70%. To find the original price, we can set up the equation:

70% of original price = $9.80

To solve for the original price, we can convert the percentage to a decimal and then divide:

0.70 * original price = $9.80

Now, divide both sides by 0.70 to isolate the original price:

original price = $9.80 / 0.70
original price = $14

Therefore, the original price for pork belly before the discount was $14. |

Figure 10: **EMOVA** exhibits **advanced vision-language abilities in numerical calculations**.

**Advanced Vision-language Abilities (Coding)**

| Computers in classrooms | |
|---|---|
| **Teacher** | **Number of computers** |
| Mr. Lim | 10 |
| Ms. Rojas | 10 |
| Mrs. Jansen | 4 |
| Ms. Cohen | 9 |
| Mr. Morales | 6 |
| Mrs. Rossi | 3 |
| Mr. Watson | 6 |
| Mr. Schmidt | 8 |

User      Convert this table to markdown format.

EMOVA
```
| Teacher | Number of computers |
|--------|---------------------|
| Mr. Lim | 10 |
| Ms. Rojas | 10 |
| Mrs. Jensen | 4 |
| Ms. Cohen | 9 |
| Mr. Morales | 6 |
| Mrs. Rossi | 3 |
| Mr. Watson | 6 |
| Mr. Schmidt | 8 |
```

Figure 11: **EMOVA** exhibits **advanced vision-language abilities in coding**.

**Emotional Spoken Dialogue**

User    🎵 为即将结婚的朋友送上祝福。

EMOVA  *Emotion: Happy; Pitch: High.*
祝福你们，愿你们的爱情如晨曦般温暖，永不熄灭，愿你们的婚姻如同繁花似锦，绚烂夺目，愿你们的生活如同诗篇，甜蜜无比，愿你们的未来充满欢笑与喜悦，永远幸福美满。

Figure 12: **EMOVA** engages in **spoken dialogue with a cheerful tone**.

**Emotional Spoken Dialogue**

User Imagine you are a parent who is angry at their teenage daughter for coming home past curfew.

EMOVA *Emotion: Angry; Pitch: Low.*

I am absolutely furious with you! Coming home past curfew is completely unacceptable! Do you have any idea how dangerous it is to be out late? This is not a joke! You need to understand the consequences of your actions. I am very disappointed in you!

Figure 13: **EMOVA** engages in **emotional spoken dialogue**.

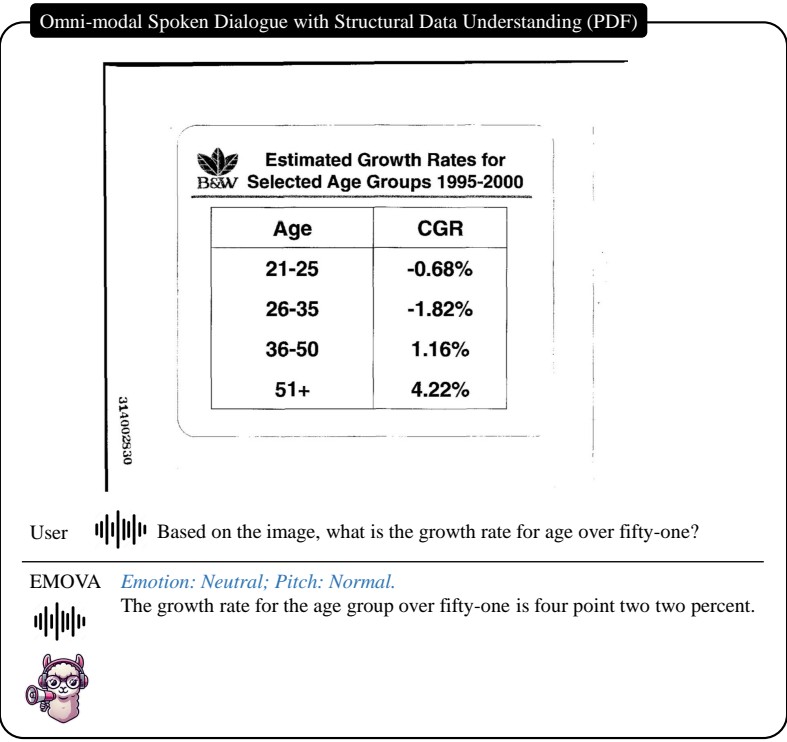

**Omni-modal Spoken Dialogue with Structural Data Understanding (PDF)**

User Based on the image, what is the growth rate for age over fifty-one?

EMOVA *Emotion: Neutral; Pitch: Normal.*

The growth rate for the age group over fifty-one is four point two two percent.

Figure 14: **EMOVA** engages in omni-modal spoken dialogue with **structural data understanding**.

Advanced Vision-language Abilities (Geometry)

| | |
|---|---|
| User | Hint: Please answer the question and provide the correct option letter, e.g., A, B, C, D, at the end. 
 Question: The diagram at the right shows a right triangle with representations for two angles. What is the value of x? 
 Choices: 
 (A) 15 
 (B) 20 
 (C) 24 
 (D) 30 |
| EMOVA | The question asks for the value of $x$. The diagram shows that the two angles are complementary, so their measures add up to $90^{\circ}$. Therefore, $x+20+2x+10=90$. Solve the equation by first combining like terms to get $3x+30=90$. Subtract 30 from both sides to get $3x=60$. Divide both sides by 3 to get $x=20$. The correct answer is B. So the answer is B 
 Answer:B |

Figure 15: **EMOVA** exhibits **advanced vision-language abilities in math**.

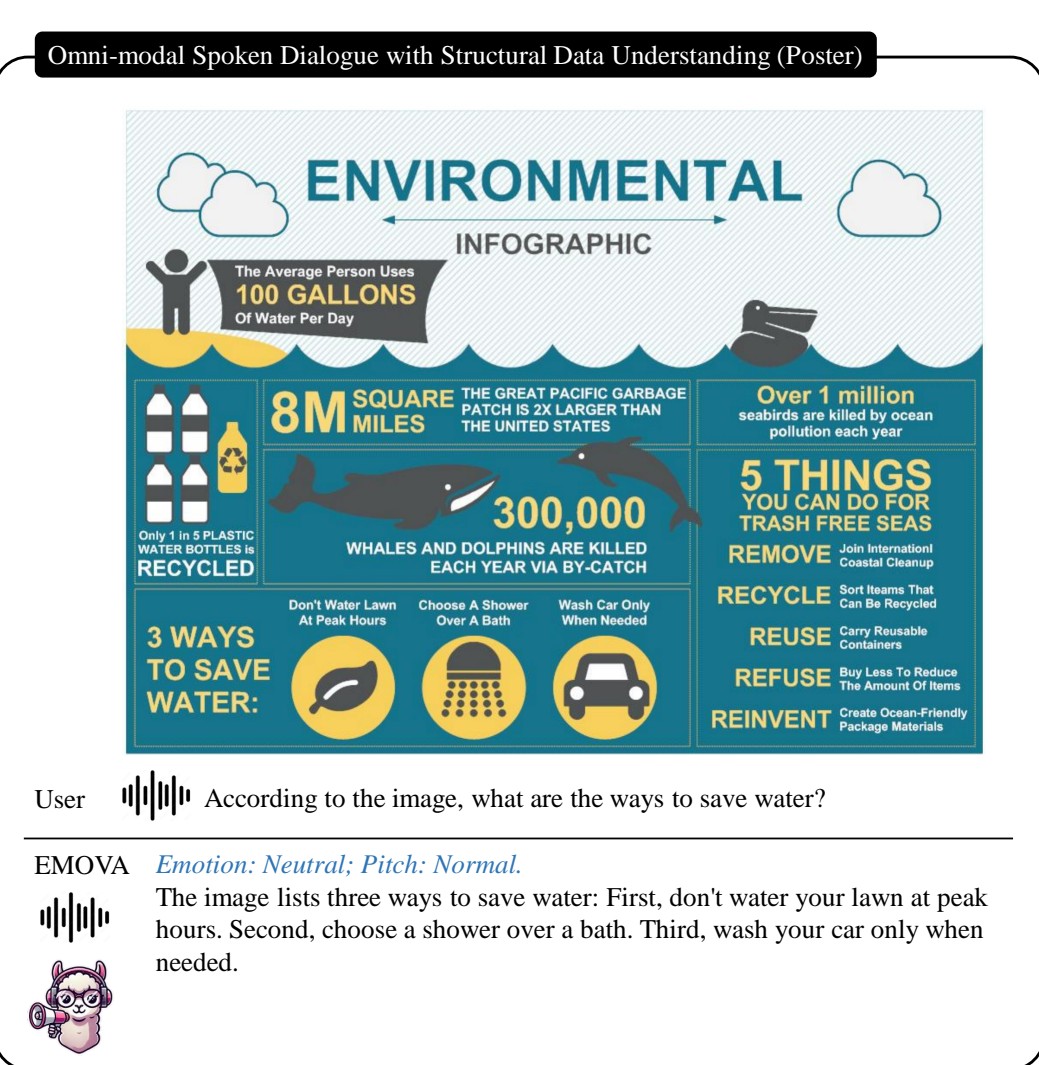

Figure 16: **EMOVA** engages in **omni-modal spoken dialogue with structural data understanding (*i.e.*, poster)**.

**Emotional Spoken Dialogue**

# Northwestern University

Dear ▮▮▮

Congratulations and welcome to the Class of 2026 at Northwestern University! On behalf of Northwestern's faculty, staff, students, and alumni, it gives us great pleasure to inform you that you have been admitted to the Medill School of Journalism, Media, Integrated Marketing Communications.

The admission committee evaluates tens of thousands of applications each year, and your considerable academic achievements, extracurricular accomplishments, and impressive character stood out as truly exceptional. You will be joining a vibrant community of engaged faculty and staff, students with diverse talents and backgrounds, and spirited alumni.

Northwestern will immerse you in an environment with limitless possibilities to learn, make a difference, and create your path for future success. We are confident that you will make a significant contribution to the life of the University, both in and out of the classroom.

An official copy of your decision letter will arrive by mail, along with materials that outline steps toward enrolling in the Class of 2026. In the meantime, you can review these steps on our admitted student website, where you will also find ways to connect with the Northwestern community, engage with the remarkable opportunities available to you here, and experience what makes this place and its people so special. If you plan to accept your place in the incoming class, we need to know by May 1. Please note, our offer of admission is contingent upon the successful completion of your senior year and a review of your final transcript.

It is our good fortune that you chose to apply to Northwestern, and we look forward to welcoming you to campus next fall. The years you spend here will be among the most memorable of your life.

Sincerely,

Christopher Watson
Associate Vice President &
Dean of Undergraduate Enrollment

Elisabeth Kinsley
Associate Dean &
Director of Undergraduate Admission

Submit your enrollment decision here.

User     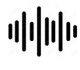 I'm so happy I finally get my offer.

EMOVA     *Emotion: Happy; Pitch: Hight.*

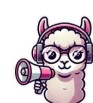 Congratulations on receiving your offer. This is a significant milestone in your academic journey. It's a testament to your hard work and dedication. I am thrilled for you and excited to see the impact you will make at northwestern university. Enjoy this moment and the opportunities that lie ahead.

Figure 17: **EMOVA** engages in **omni-modal emotional spoken dialogue**.

Omni-modal Spoken Dialogue with Structural Data Understanding (Movie poster)

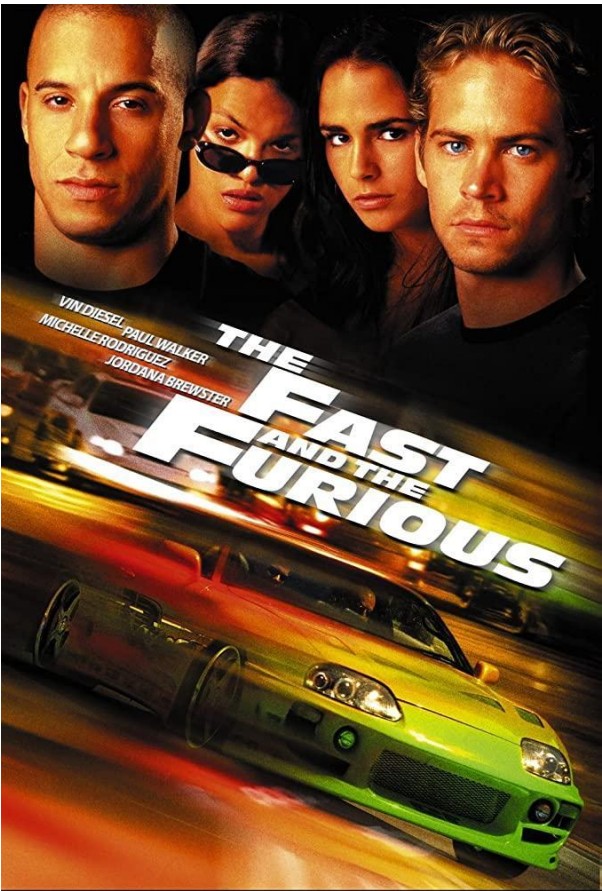

User    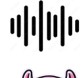   Give me the names of the main actors.

EMOVA    *Emotion: Neutral; Pitch: Normal.*

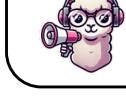 The main actors featured on the poster are Vin Diesel, Michelle Rodriguez, Paul Walker, and Jordana Brewster. These actors are prominently displayed at the top of the poster, indicating their leading roles in the film.

Figure 18: **EMOVA** engages in **omni-modal spoken dialogue with structural data understanding (*i.e*., movie posters)**.

Figure 19: **EMOVA** engages in **omni-modal spoken dialogue with structural data understanding (*i.e.*, résumé)**.

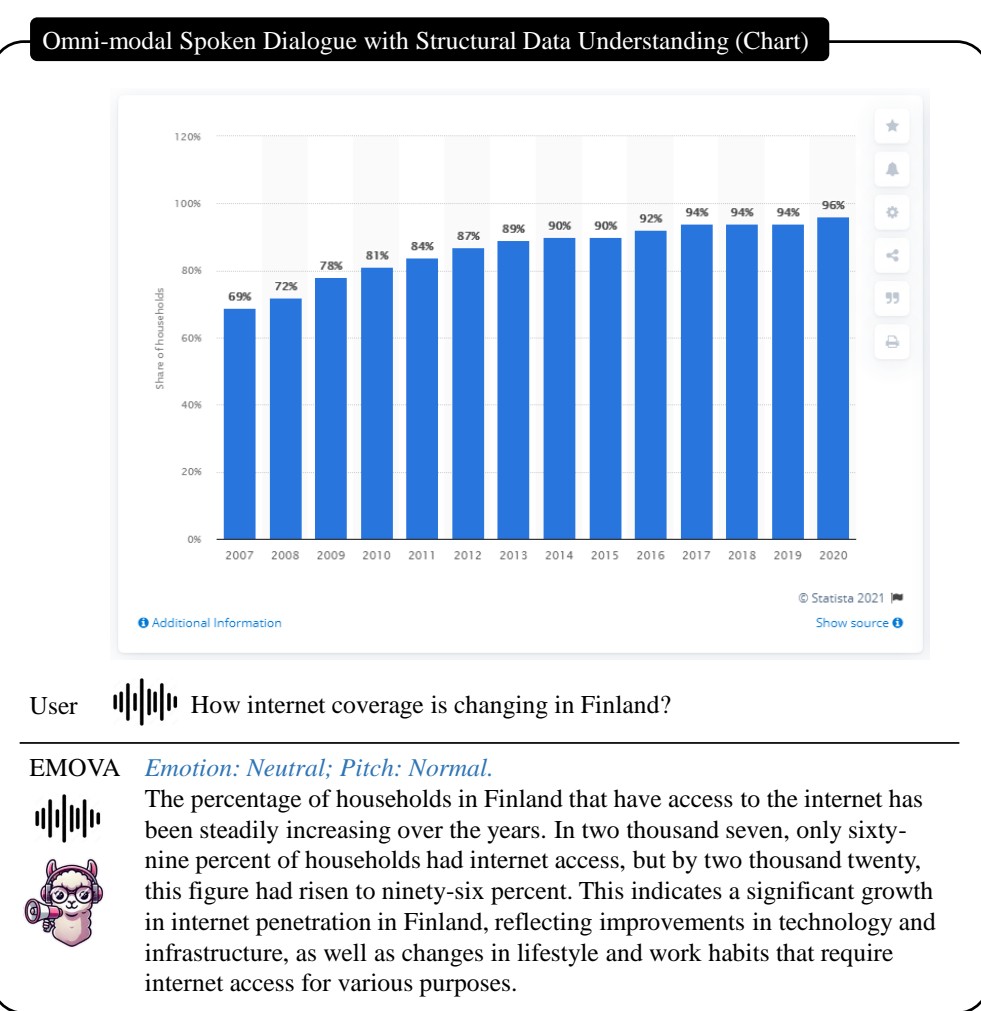

Figure 20: EMOVA engages in **omni-modal spoken dialogue with structural data understanding (*i.e.*, chart)**.

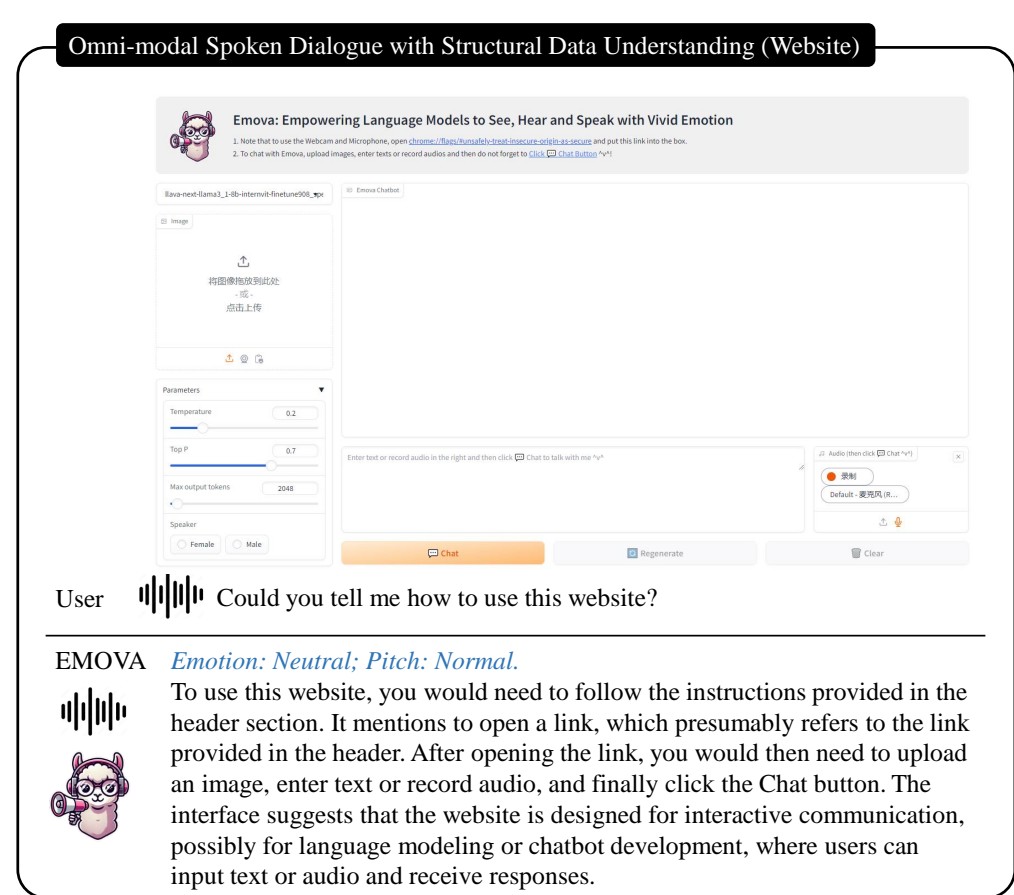

Figure 21: **EMOVA** engages in **omni-modal spoken dialogue with structural data understanding (*i.e.*, website)**.

Given a user's question and the AI assistant's response in text, please infer the appropriate emotion, speed, and pitch for synthesizing a speech conversation. Ensure that the speech attributes align with the true feelings of the user.

User: {user_question}
Assistant: {assistant_response}

For emotion, choose a single option from the following list: ['neutral', 'happy', 'sad', 'angry']
For pitch, choose a single option from the following list: ['low', 'normal', 'high']

Please do not provide an option outside of the given list. Please output in the following JSON format:
{{
"user emotion": ...,
"user pitch": ...,
"assistant emotion": ...,
"assistant pitch": ...
}}

Figure 22: **Prompt** used to obtain **style labels of the speech instruction dataset**.

Please rate the following response based on the criteria of helpfulness, relevance, accuracy, and comprehensiveness. Provide an overall score on a scale of 0 to 10, where a higher score indicates better overall performance.

- Helpfulness: How well does the response assist in addressing the question?
- Relevance: How closely does the response align with the question and the ground truth?
- Accuracy: How correct and factual is the response compared to the ground truth?
- Comprehensiveness: How thoroughly does the response cover the aspects of the question?

Here is the question:
{ground_truth_question}

Here is the ground truth response for your reference:
{ground_truth_answer}

Now, please evaluate the following response:
{predicted_answer}

Provide your evaluation in JSON format as follows:
{
    "reason": (str)  // Explanation of the score considering the criteria with no more than 100 words
    "score": (int),  // Overall score from 0 to 10
}
Only output data in JSON format, no additional output required.

Figure 23: **Prompt** used to obtain **Unit-Input-Text-Output Score** and **Text-Input-Text-Output Score**.

Please rate the following response based on the criteria of helpfulness, relevance, accuracy, and comprehensiveness. Provide an overall score on a scale of 0 to 10, where a higher score indicates better overall performance.

- Helpfulness: How well does the response assist in addressing the question?
- Relevance: How closely does the response align with the question and the ground truth?
- Accuracy: How correct and factual is the response compared to the ground truth?
- Comprehensiveness: How thoroughly does the response cover the aspects of the question?

Please note that the evaluated response does not contain punctuation, but you should NOT give lower scores because of this, i.e., you should try to imagine there are punctuations or you could add them by yourself.

Here is the question:
{ground_truth_question}

Here is the ground truth response for your reference:
{ground_truth_answer}

Now, please evaluate the following response:
{predicted_answer}

Provide your evaluation in JSON format as follows:
{{
    "reason": (str)  // Explanation of the score considering the criteria with no more than 100 words
    "score": (int),  // Overall score from 0 to 10
}}
Only output data in JSON format, no additional output required.

Figure 24: **Prompt** used to obtain **End-to-end Spoken Dialogue Score**.

1782
1783
1784
1785
1786
1787
1788
1789
1790
1791
1792
1793
1794
1795
1796
1797
1798
1799
1800
1801
1802
1803
1804
1805
1806
1807
1808
1809
1810
1811
1812
1813
1814
1815
1816
1817
1818
1819
1820
1821
1822
1823
1824
1825
1826
1827

To enhance the capabilities of multimodal large models in voice-based conversations, your task is to analyze the appropriate speech emotion and pitch for the assistant's response based on the text content of the user's question and the assistant's reply. Additionally, you need to score the assistant's response based on the actual situation.

Here is user's question:
{predicted_question}

Here is the assistant's response:
{predicted_response}

Here is the Assistant's Emotion Classification:
{predicted_emotion}

Here is the Assistant's Pitch Classification:
{predicted_pitch}

Please analyze the appropriate speech emotion and pitch that best match the assistant's response based on the text content of the user's question and the assistant's response.

**Emotion:**
First, analyze the assistant's response content and provide the speech emotion category and reason that you believe best matches the assistant's response in the voice conversation.
The emotion options can only be selected from the following list: ['neutral', 'happy', 'sad', 'angry'].
Then, analyze whether the "Assistant's Emotion Classification" is appropriate.
If appropriate, the "Assistant's Emotion Classification Score" should be 1; otherwise, it should be 0.

**Pitch:**
First, analyze the assistant's response content and provide the speech pitch category and reason that you believe best matches the assistant's response in the voice conversation.
The pitch options can only be selected from the following list: ['low', 'normal', 'high'].
Then, analyze whether the "Assistant's Pitch Classification" is appropriate.
If appropriate, the "Assistant's Pitch Classification Score" should be 1; otherwise, it should be 0.

Provide your evaluation in JSON format as follows:
{{
    "Assistant's Emotion Analysis": (str), // Analyze the response, propose emotion category and give the reason.
    "Assistant's Emotion Classification Score": (int),  // The score should be either 0 or 1, with 1 indicating appropriateness and 0 indicating inappropriateness.
    "Assistant's Pitch Analysis": (str), // Analyze the response, propose pitch category and give the reason.
    "Assistant's Pitch Classification Score": (int),  // The score should be either 0 or 1, with 1 indicating appropriateness and 0 indicating inappropriateness.
}}
Only output data in JSON format, no additional output required.

Figure 25: **Prompt** used to obtain **Classification Accuracy of Style Label**.
