# OpenReview forum: "EMOVA: Empowering Language Models to See, Hear and Speak with Vivid Emotions"
_ICLR.cc/2025/Conference — ICLR 2025 Conference Withdrawn Submission_

### Official Review · Reviewer_zGnp · 2024-10-18

**Soundness:** 2
**Presentation:** 4
**Contribution:** 3
**Rating:** 6
**Confidence:** 4

**Summary:**

This paper proposes a novel system that supports empathetic interaction with humans via speech as well as being able to understand images. The authors propose to disentangle the speech content and speech style into different embeddings. To achieve a better modality alignment among text, speech and image, the authors propose the omni-modality text-centric alignment using the proposed EMOVA system to train on bi-modal alignment data simultaneously.

**Strengths:**

1. The presentation of the paper is very good. It is easy to follow with clear illustrative figures and plots. The equation notations are also clear and accurate.
2. The proposed EMOVA system contributes to the research community as it is the first single neural network that integrates visual ability into a speech-in-speech-out system and supports speech synthesis with emotions.
3. The separation of semantic and style tokens is a new application of the existing disentanglement study in the TTS field to the speech-language model research field and experiments have demonstrated the usefulness of this design.

**Weaknesses:**

1. The core or central motivation of this paper is a bit unclear to me. Is the motivation to include visual ability in a speech-language model (which is not reflected in its name EMOVA), or to include speech synthesis with emotion in a speech-language model? I agree with the authors that the system is the first to achieve these functionalities in a single system, but I am not sure how those functionalities are connected to be put in a single paper. Why being able to understand an image is crucial to emotional speech synthesis, or why emotional speech synthesis is crucial for understanding an image? Taking the example the authors show, "being able to write a sad poem about the image" and "being able to synthesize sad speech about the image" are not strongly connected in my opinion. Note that this problem also exists in the experiments since none of the test sets used actually reflect the importance of __jointly__ understanding image and interacting in speech with emotions, apart from case studies.
2. Regarding novelty, which is also related to motivation, I need more clarification on which parts of the system haven't been done before. For example, there are full-duplex systems [1,2] that can understand speech and output speech. There exists a lot of work in the TTS community working on disentangling style and text representations for speech synthesis [3,4]. There is work controlling emotion in TTS using natural language prompts [5]. There also exist systems that can understand both speech and visual inputs [6]. I understand that this is the first single speech-LM that achieves all the functionalities above (which I gave credit to in the strength part), but the actual approach for each functionality does not seem to be novel to me.
3. The experiments showed that EMOVA outperforms GPT-4v on various benchmarks, but most of them are in the training set (hence not zero-shot).

[1]. https://arxiv.org/pdf/2408.02622
[2]. https://arxiv.org/abs/2409.06666
[3]. https://arxiv.org/abs/2306.07691
[4]. https://arxiv.org/abs/1810.07217
[5]. https://arxiv.org/abs/2406.06406
[6]. https://arxiv.org/pdf/2406.15704

**Questions:**

1. For experiments with GPT-4v and Gemini, Is this comparison fair? Is it possible to test EMOVA's zero-shot ability on some unseen task and show that it outperforms GPT-4v?
2. Where can I find the performance comparison for the synthesis TTS quality? How does it compare to a cascaded system where you have a separate TTS component which takes speech content and style descriptions as input, and EMOVA just generate the text response and then generates a description of how to speak that response out? If there is little difference, then why do we need a full duplex model?
3. Why is the emotion in Fig. 5 useful? If I were to prompt a system to write a poem about an image (which is itself a weird task though), why does reading it with a sad voice better than with a plain voice if I am not instructing it to say it that way?

---

### Official Review · Reviewer_S2KR · 2024-10-29

**Soundness:** 2
**Presentation:** 2
**Contribution:** 2
**Rating:** 5
**Confidence:** 4

**Summary:**

This paper presents a multi-modal LLM with the ability to control the style of output speech.

**Strengths:**

1. The motivation of this study is clear which is to build a multi-modal LLM with the ability to control the style of output speech.
2. A new model architecture and corresponding model training strategies are designed.
3. The proposed method has been evaluated using several vision-language and speech benchmarks.

**Weaknesses:**

1. The descriptioins of the proposed method are not clear. For example, as the authors mentioned, the semantic-acoustic disentangled speech tokenizer is one of the contributions of this paper. However, the introduction to the disentanglement in Section 3.2 is very simple and it is difficult for readers to get the key ideas.
2. The experimental results are not convincing enough. Only the overal performances of the proposed model on different tasks were evaluated. There are no ablation studeis on the key modules of the proposed method which makes it difficult to accept the contributions of the proposed method. In Section 5.3, there were no comparative models for evaluating the emotion-rich spoken dialogue, and only the synthetic data was used for evaluation.

**Questions:**

According to Section 4.2, the authors define "style" as a broader concept than "emotion". So, only mentioning "emotion" in the paper title and  most descriptions may not be appropriate.

---

### Official Review · Reviewer_3dLu · 2024-11-04

**Soundness:** 3
**Presentation:** 4
**Contribution:** 3
**Rating:** 5
**Confidence:** 3

**Summary:**

The paper presents EMOVA (Emotionally Omni-present Voice Assistant), a novel end-to-end, omni-modal language model with the capability to understand three modalities: vision, language, and speech and generate speech with vivid emotion. EMOVA uses a continuous vision encoder and a semantic-acoustic disentangled speech tokenizer for seamless omni-modal alignment and diverse speech style controllability. An efficient text-centric omni-modal alignment is introduced to further improve the vision-language and speech capabilities.

**Strengths:**

1. EMOVA model has the ability to understand vision, speech and text modalities. Through the introduced text-centric omni-modal alignment, EMOVA maintains a comparable performance on vision-language benchmarks.
2. Integrating LLM with a semantic-acoustic distengled speech tokenizer and Unit-to-speech detokenizer with style control built on VITS, EMOVA can generate speech in different styles.

**Weaknesses:**

1. Since only the semantic embedding of speech is quantified to speech units, due to the semantic-acoustic distengled modeling of speech tokenizer, the emotional information in the input speech is not conveyed to the LLM. The model generates emotion labels only based on vision and text modalties, which is not often the case in real dialogue.
2. On vision-language benchmarks, SOTA VLLMs are compared. While on speech benchmarks, only omni-modal LLMs are compared. SOTA SLLMs such as Qwen-Audio and SALMONN are not compared. So the conclusion "EMOVA obtains state-of-the-art performance on speech benchmarks" is doubtful.

**Questions:**

1. For the training paradigm, there is a vision-language pre-alignment before omni-modal text-centric alignment. Since in section 4, the observation "image-text and speech-unit-text data benefit each other" is derived, I wonder what will happen if we skip the pre-alignment stage and start with text-centric omni-modal alignment.
2. Since emotional speech is generated by TTS systems, I wonder if the synthesized speech can convey emotion in a natural way like a human.
3. There are two output modes of EMOVA, speech or text. Does the model choose the output modality by itself or the output modality is specified in the system prompt?  As illustrated in Figure 3, when generating speech, the outputs are formatted as JSON. I wonder whether forcing the model to output JSON format when generating speech will harm the performance.

---

### Official Review · Reviewer_VbJh · 2024-11-05

**Soundness:** 3
**Presentation:** 2
**Contribution:** 3
**Rating:** 5
**Confidence:** 4

**Summary:**

The paper  presents EMOVA, a multimodal LLM with  image and speech understanding and controllable speech synthesis capabilities (speaker and emotion). The model is based on LLama 3.1 8B and tuned for speech-language and image-language tasks using a 3-stage pipeline, i.e., tuning of the projection layer, alignment, and SFT. The key observation of this paper is that using bi-modal datasets (speech-text and vision-text) with the text as the central modality is beneficial for both vision-language specific and speech specific tasks, aleviating the need for instructions containing all three modalities. Furthermore, disentanglement of the speech representations into semantic and style components improves results and allows for controllability of the synthetic speech. The model achieves sota results in vision-language and speech benchmarks.

**Strengths:**

The model presented in the paper is a step forward for multimodal language models. The key conclusion, i.e., that training with mixed bi-modal instruction pairs, along with the observation of mutual benefit of the visual and speech modalities, can allow future researchers to extend the multimodal capabilities of LLMs to more diverse modalities.

Regarding specifics, I find the contribution on the speech pipeline with the semantic and style disentanglement of the speech representations especially interesting. Furthermore, I appreciate the  breakdown of the instruction sets used for fine-tuning and the evaluation presented in Figure 4.

**Weaknesses:**

The key negative of this paper is the presentation, which I find a bit cramped. The presentation of key results, intuition and the relation to the literature is good, but the description of the key parts of the architecture (speech tokenizer / detokenizer) and the experimental procedure is delegated into the appendix, calling into question the rigor in the main part of the paper. Also, I find the constant referencing of upcoming sections a bit tedious for the reader.

Secondly, I find the lack of human evaluation for the speech synthesis as an omission. The examples presented in the example page are impressive, but the voice sounds robotic, which calls into question whether the end to end approach is practical in this scenario, or having a separate TTS module which can be independently optimized would be preferable.

Finally, some key information is missing, i.e. whether the model will be released as open-source (and the relevant license) and the amount of compute used for training (what kind of GPUs, how many, for how long). I find this information to be essential when presenting a new model to the community.

**Questions:**

* I would appreciate the authors comments regarding the practicality of adopting a fully end-to-end approach from the viewpoint of improving separate submodules and extending the system capabilities (e.g., to other languages).
* Is the use of gpt-4o for the creation of training labels in compliance with the OpenAI terms of use?

**Details Of Ethics Concerns:**

L359-L360: The style labels are obtained by prompting GPT-4o to make reasonable inferences given the dialogue context.

The use of GPT-4o to produce training data should be investigated for compliance reasons. The relevant page (https://openai.com/policies/row-terms-of-use/) states that it is against the terms of use to "Use Output to develop models that compete with OpenAI.". Whether this applies in this case is subject to investigation

---

### Note · Authors · 2024-11-15

I have read and agree with the venue's withdrawal policy on behalf of myself and my co-authors.